# Comparative Proteomics Unveils LRRFIP1 as a New Player in the DAPK1 Interactome of Neurons Exposed to Oxygen and Glucose Deprivation

**DOI:** 10.3390/antiox9121202

**Published:** 2020-11-30

**Authors:** Núria DeGregorio-Rocasolano, Verónica Guirao, Jovita Ponce, Marc Melià-Sorolla, Alicia Aliena-Valero, Alexia García-Serran, Juan B. Salom, Antoni Dávalos, Octavi Martí-Sistac, Teresa Gasull

**Affiliations:** 1Cellular and Molecular Neurobiology Research Group, Department of Neurosciences, Germans Trias i Pujol Research Institute, 08916 Badalona, Catalonia, Spain; ndgregorio@igtp.cat (N.D.-R.); vguirao@igtp.cat (V.G.); jovitaponce@gmail.com (J.P.); mmelia@igtp.cat (M.M.-S.); agarcias@igtp.cat (A.G.-S.); 2Unidad Mixta de Investigación Cerebrovascular, Instituto de Investigación Sanitaria La Fe—Universidad de Valencia, 46026 Valencia, Spain; a.aliena.v@gmail.com (A.A.-V.); salom_jba@gva.es (J.B.S.); 3Departamento de Fisiología, Universidad de Valencia, 46010 Valencia, Spain; 4Neurosciences Department, Hospital Germans Trias i Pujol, 08916 Badalona, Catalonia, Spain; adavalos.germanstrias@gencat.cat; 5Department of Cellular Biology, Physiology and Immunology, Universitat Autonòma de Barcelona, 08193 Bellaterra, Catalonia, Spain; 6Fundació Institut d’Investigació en Ciències de la Salut Germans Trias i Pujol (IGTP), Carretera del Canyet, Camí de les Escoles s/n, Edifici Mar, 08916 Badalona, Catalonia, Spain

**Keywords:** LRRFIP1, DAPK1, neuron, OGD, MCAO, NMDA, ROS, ferroptosis

## Abstract

Death-associated protein kinase 1 (DAPK1) is a pleiotropic hub of a number of networked distributed intracellular processes. Among them, DAPK1 is known to interact with the excitotoxicity driver NMDA receptor (NMDAR), and in sudden pathophysiological conditions of the brain, e.g., stroke, several lines of evidence link DAPK1 with the transduction of glutamate-induced events that determine neuronal fate. In turn, DAPK1 expression and activity are known to be affected by the redox status of the cell. To delineate specific and differential neuronal DAPK1 interactors in stroke-like conditions in vitro, we exposed primary cultures of rat cortical neurons to oxygen/glucose deprivation (OGD), a condition that increases reactive oxygen species (ROS) and lipid peroxides. OGD or control samples were co-immunoprecipitated separately, trypsin-digested, and proteins in the interactome identified by high-resolution LC-MS/MS. Data were processed and curated using bioinformatics tools. OGD increased total DAPK1 protein levels, cleavage into shorter isoforms, and dephosphorylation to render the active DAPK1 form. The DAPK1 interactome comprises some 600 proteins, mostly involving binding, catalytic and structural molecular functions. OGD up-regulated 190 and down-regulated 192 candidate DAPK1-interacting proteins. Some differentially up-regulated interactors related to NMDAR were validated by WB. In addition, a novel differential DAPK1 partner, LRRFIP1, was further confirmed by reverse Co-IP. Furthermore, LRRFIP1 levels were increased by pro-oxidant conditions such as ODG or the ferroptosis inducer erastin. The present study identifies novel partners of DAPK1, such as LRRFIP1, which are suitable as targets for neuroprotection.

## 1. Introduction

Modern biological research reveals that, in order to regulate essential cellular functions in a given condition, the cell sets its proteome into highly complex assemblies of multiproteins. In this regard, death-associated protein kinase 1 (DAPK1) is an emerging hub kinase involved in a number of cellular functions, both in physiological and pathophysiological conditions. While it has been associated to excitotoxic neuronal damage, its role in ischemia is still poorly studied.

A relationship between reactive oxygen species (ROS) and DAPK1 has been observed. Thus, in tumor cell lines, it has been described that antioxidants increase the expression of DAPK1 [1], and that ROS facilitate protein phosphatase 2A (PP2A)-mediated dephosphorylation of pDAPK1 to render the active, non-phosphorylated, DAPK1 form [2]. In the brain, it has been recently reported that ischemia downregulates miR-98-5p, and that experimentally upregulated miR-98-5p in stroked mice inhibits ROS production, reduces infarction and suppresses DAPK1 signaling [3].

During ischemia, the excess glutamate released by presynaptic glutamatergic terminals spills over the synaptic cleft to interact with and signal downstream of the extrasynaptic *N*-methyl-d-aspartate (NMDA) receptor (NMDAR) and also increases the presence of ROS-modified lipids in the cell membrane. NMDAR at the neuronal membrane is pivotal for normal cell activity and neurotransmission, but its overactivation plays a principal role in neurodegeneration through excitotoxicity. As direct antagonism of NMDAR has been revealed to be poorly tolerated in the clinical arena so far, other therapeutic strategies have been postulated to prevent the association of the NMDAR with intracellular protein signaling transducers networking for excitotoxicity. Specifically, the cytoplasmic C-terminal domain (CTD) of the NMDAR 2B subunit (NR2B), which in neurons is located mostly extrasynaptically, has been proposed as a good therapeutic target for this purpose, based on the fact that CTD directly senses Ca^2+^ entry through NMDAR. Replacing the CTD of NR2B with that of 2A subunit-containing NMDAR (NR2A) by targeted exon exchange has been shown to reduce vulnerability to excitotoxicity [4]. Additionally, NR2B-containing NMDAR have been identified, and recently retested and confirmed, to be primary mediators of ischemic stroke damage [5,6]. Therefore, although other intracellular mechanisms are involved in excitotoxicity, CTD-coupled events are thought to be the first downstream excitotoxicity intracellular signals towards cell death.

While the mechanisms of CTD-NR2B-mediated excitotoxicity still remain largely unexplained, it is worth highlighting that: (1) the CTD-postsynaptic density (PSD) 95-neuronal nitric oxide synthase (nNOS) or CTD-DAPK1 hubs are involved in excitotoxicity, as shown by protection exerted by genetic knockdowns [7,8]; (2) very recently, other authors identified antagonistic effects of ischemia-regulated endogenous miRNAs (AK038897 and miR-26a-5p) on DAPK1 that finally affect cerebral ischemia/reperfusion injury [9]; and (3) mice carrying a specific mutation that prevents binding of DAPK1 to NR2B are protected against stroke damage by inhibiting injurious Ca^2+^ influx through extrasynaptic NMDAR channels without altering synaptic NMDAR functions [5]. While the absence of DAPK1 has been shown to protect neurons from a variety of acute insults, the true requirement of DAPK1 for the excitotoxic signaling to proceed has also been questioned [10].

Some pioneering studies revealed that DAPK1 is the most prevalent protein recruited to the cytoplasmic tail of NR2B during cerebral ischemia [8], with virtually no interaction with NR2A. During cerebral ischemia, DAPK1 is dephosphorylated and then interacts and phosphorylates the CTD-NR2B to enhance the inflow of Ca^2+^ into the neuron, further exacerbating excitotoxicity in a vicious circle [11]. The fact that this effect of DAPK1 on the increase of Ca^2+^ entry has not been observed through synaptic, mostly NR2A, NMDAR [8,12] reinforces the idea that the reciprocal DAPK1 interaction with NR2B is specific and differential, and implies detrimental, if not dramatic, functional consequences for the neuron.

Despite the growing role of DAPK1 as an intracellular hub kinase, the mediators through which NR2B-DAPK1 complex leads towards neuronal death in stroke remain unclear. Therefore, increasing the knowledge of DAPK1 interactions of the neuron in ischemic and ischemic-like conditions is a sine qua non condition to broaden the array of molecules susceptible to further functional studies as a strategy in the search for therapies in cerebral ischemia. Here, we aimed to improve the knowledge of these mediators, and especially the protein DAPK1 interactors that are differentially increased, by chasing them together with DAPK1 and subjecting them to high-resolution proteomics to get a general picture of the DAPK1 interactome of the neuron under oxygen and glucose deprivation, the most frequently used in vitro model of ischemic stroke.

## 2. Materials and Methods

### 2.1. Primary Culture of Cortical Neurons

The experimental protocols used to obtain neurons were approved by the Institutional Animal Care and Animal Experimentation Committee of the IGTP and the Government of Catalonia (DAAM5876) and were conducted according to international guidelines. A total of 21 pregnant rats weighing 200–250 g (8 to 15 fetuses/pregnant rat) were used in this part of the study. Animals were housed individually in standard conditions of temperature (22 °C) and photoperiod (12-h light/dark cycle) with food and water being ad libitum. For each experimental condition, we used at least three biological replicates (independent primary neuron culture preparations) performed using neurons obtained from fetuses from different pregnant rats and were conducted on different days. For each biological replicate, three to four technical replicates (culture plate wells) were made. We adhered to and followed the 3Rs principle in animal research to minimize animal distress and to decrease the number of animals used. Experiments were carried out between 08 h and 20 h.

In brief, pregnant rats we anesthetized with 4% isoflurane in a 30% O_2_/70% N_2_O mixture, fetuses rapidly and carefully excised for further processing, and rats euthanized by decapitation. Primary cultures of rat cortical neurons were prepared, as we previously described [13,14], from E18 fetuses obtained from pregnant Sprague-Dawley rats (Envigo/Harlan, Barcelona, Spain). Neurons in culture were grown in Neurobasal™ medium (Gibco, Alcobendas, Spain) supplemented with 2% B-27 (Life Technologies, Alcobendas, Spain), 0.5 μM L-glutamine and 40 μg/mL gentamicin and kept in an incubator (Galaxy RX, RS Biotech, Irvine, UK) at 37 °C in 95% air with 5% CO_2_. Medium was changed following the supplier instructions, and the experiments were performed at 11–12 DIV.

### 2.2. Oxygen and Glucose Deprivation (OGD)

The culture medium in which the neurons had matured (conditioned medium) was removed and stored. Neurons exposed to OGD were incubated for 90 min in glucose-free DMEM in a 2% O_2_ atmosphere in a Galaxy RX hypoxia incubator (RS Biotech). Control cells were incubated for 90 min in DMEM supplemented with 4.5 g/L glucose at normoxia (20% O_2_). After this 90 min incubation period, cultures were returned to their conditioned medium and further incubated with 4.5 g/L glucose in normoxic conditions. Proteins for WB or immunoprecipitation (IP) procedures were obtained 30 min after returning to normal oxygen and glucose conditions.

### 2.3. Treatments

Neurons in their conditioned medium were incubated with NMDA (50 µM) in the presence of 10 µM glycine for 30 min before harvesting the protein or proceed to immunocytochemistry (ICC). Neurons in their conditioned medium were exposed to erastin (MCE/Quimigen, 10 or 20 µM dissolved in 0.2% DMSO) for up to 72 h.

### 2.4. Assessment of Neuronal Viability

Cell death by OGD or NMDA was determined by measuring the incorporation of propidium iodide along 24 h. We used a Varioskan flash reader (Thermo Fisher Scientific, Alcobendas, Spain) and followed the method by Rudolph and col. [15], adapted in our lab [13,14]. Ferroptosis induced by erastin produced delayed neuronal death that was assessed by measuring the conversion of 3-(4,5-dimethylthiazol-2-yl)-2,5-diphenyltetrazolium bromide (MTT) to formazan as a result of metabolic activity in living cells.

### 2.5. Co-Immunoprecipitation (Co-IP) Procedure to Identify Protein Interactors by LC-MS/MS

For immunoprecipitation of intact protein complexes to be used in LC-MS we used Dynabeads (Db) Co-Immunoprecipitation Kit (Thermo Fisher Scientific), which is based on the covalent binding of a specific antibody on the surface of magnetic beads, easily separable from the suspension by the use of a dedicated magnet. Following the supplier recommendations for an antibody (Ab) in a sodium azide solution, 8 μg of the specific DAPK1 antibody (DAPK1-Ab raised in rabbit) was covalently attached to each mg of Db (DAPK1-Ab-Db), unbound Ab was extensively washed, and DAPK1-Ab-Db were incubated with 50 mg protein extracts obtained from 90 × 106 pure cortical neurons either exposed to the control or OGD. After extensive washing, DAPK1 recognized by the specific Ab along with any other proteins or ligands bound to DAPK1 in each condition were recovered in the kit elution buffer to be used in WB procedures or in 1% tricarboxylic acid (TCA) solution to be used in high-resolution liquid chromatography tandem mass spectrometry (LC-MS/MS). Negative controls of IP were performed with a non-specific antibody. To maximize consistency, volumes, protein concentrations, and the amount of antibody used were kept constant in our IP-LC-MS/MS experiment. Samples for LC-MS/MS were digested using filter-aided sample preparation [16] and trypsin as an enzyme, following the internal protocols of the ‘Laboratori de Proteòmica-CSIC/Universitat Autònoma de Barcelona’. The samples were injected in a chromatographic system equipped with a C18 preconcentration column (300 μm id × 0.5 cm) and an analytical column (150 μm id × 15 cm). Peptides were loaded onto the preconcentration column using 1% formic acid (FA) as solvent and eluted directly to the analytical column with a flow of 400 nL/min using a gradient of 0–40% acetonitrile in 0.1% FA for 120 min. The chromatographic system was connected on-line to a high-resolution mass spectrometer LTQ-Orbitrap XL (Thermo Fisher Scientific) performing the analysis in dependent sweep mode: a complete sweep using the Orbitrap with a resolution of 60,000 and 10 parallel sweeps of MS/MS in the ion trap on the most abundant precursors. An exclusion time of 30 s was included to avoid repetitive analysis of dominant signals. Identification of the peptides in the database was carried out using the Proteome Discoverer™ Software (Thermo Fischer Scientific) using the Uniprot database restricted to rodents and the following parameters: mass tolerance of peptide 2 Da and 20 ppm for low and high resolution, respectively, tolerance of 0.8 Da fragments, trypsin enzyme, allowing up to 2 missed junctions, dynamic modification of methionine oxidation (+16 Da) and fixed modification of cysteine carbamidomethylation (+57 Da). Identifications were filtered at a 0.1% false discovery rate, and only proteins identified with two or more peptides were considered. Peptide spectrum matches and coverage parameters provided by the search Proteome Discoverer™ Software rendering ratios altered >1.4-fold were considered proteins with a possible change in their levels in Co-IP-DAPK1 after OGD.

### 2.6. Co-Immunoprecipitation Assays

The proteins identified in the DAPK1 interactome represent a network of proteins interacting with one another as partners that act in concert in specific cellular mechanisms, not necessarily existing in all of them in a single complex at a given time and condition. Results obtained after the DAPK1 Co-IP plus LC/MS interactome approach predicted the increased presence after OGD, among others, of the interactor candidate leucine-rich repeat in flightless 1 interaction protein 1 (LRRFIP1). Thereafter, DAPK1 and LRRFIP1 were subjected to direct and reverse Co-IP, and their interaction were further confirmed by WB. Fifteen mg ischemic brain tissue, obtained 24 h post ischemia onset, or 0.4 × 106 primary cortical neurons, obtained in control conditions, or exposed to NMDA for 30 min or to OGD, were homogenized in SDS-free RIPA (RIPA-Co-IP), protein extracts were obtained and used to test Co-IP. The indirect IP procedure was initiated by incubating the same amount of the protein extract sample overnight with the specific capture antibody anti-DAPK1 (RRID:AB_259206), anti-LRRFIP1 (RRID:AB_10570968), or appropriate non-specific control antibodies. The pre-formed antibody-protein complex was incubated 10 min with PureProteome protein G Magnetic beads (PPpGMb) (Millipore), and beads were then extensively washed using magnetic pelleting between washes. The resulting PPpGMb-bound immunocomplexes were eluted by SDS and analyzed by WB according to standard techniques. Since this method using PPpGMb does not involve covalent binding to the beads, bands appearing around the 50 kDa marker in the WB of PPpGMb-based pulling down experiments might well be, in part, heavy chains of the antibody used to prey the interactome that elute together with the proteins of interest.

### 2.7. Western Blot

Twenty-five μg of total protein from brain or 10 μg of total protein from cortical neurons in culture were used to assess protein expression by WB. Proteins obtained from control, NMDA or OGD samples following IP were prepared in WB loading buffer and loaded in Precast NuPAGE Midi 10% Bis-Tris (Life Technologies). MW markers (Magic Mark XP or Novex Sharp Pre-Stained Protein Standard; Life Technologies) were included in 10% Bis-Tris gels. For protein extraction addressed to detect phosphorylated proteins, we used a lysis buffer containing 20 mM Tris-Cl pH 7.5, 137 mM NaCl, 1% Nonidet P-40, 0.5% sodium deoxycholate, 100 mM phenylmethylsulfonyl fluoride and Complete™ EDTA-free Protease Inhibitor Cocktail supplemented with 2 mM EDTA, 5 mM sodium orthovanadate and 50 mM sodium fluoride. The presence of the proteins of interest was detected and analyzed with an image generation and analysis system based on the Odyssey Near Infrared Fluorescence (Li-COR). When required, actin or tubulin were used as tissue sample loading controls and to normalize data in expression experiments. DAPK-1 band patterns obtained in WB using the two anti-DAPK1 antibodies (a polyclonal Ab against aa of the N-term DAPK1 molecule or a monoclonal clone DAPK-55 against the whole molecule) listed in the antibodies section were similar in recognizing both full-length and truncated DAPK1 forms.

### 2.8. Immunocytochemistry

Neurons grown on poly-l-lysine-coated glass coverslips were fixed at 4 °C in 4% paraformaldehyde/2% sucrose in PBS, pH 7.2. To address cellular localization, neuronal cultures were processed to detect N-terminal extracellular regions of the NR2B. Neurons were then permeabilized with 100% methanol for 8 min at −20 °C to detect intracellular epitopes (DAPK1 or LRRFIP1), blocked with 3% bovine serum albumin in PBS, and incubated overnight at 4 °C with the primary antibodies. Subsequently, the cultures were washed, incubated with secondary antibodies and mounted with Fluoromount (Sigma-Aldrich, Madrid, Spain) or ProLong^®^ Gold (Life Technologies). Neurons were imaged in a confocal microscope (LSM710; Carl Zeiss, Tres Cantos, Spain) using an immersion oil 63x lens (N.A. 1.4), a MCR5 camera and Zen Black software (Carl Zeiss).

### 2.9. Transient Right Focal Cerebral Ischemia

Transient stroke experimental protocols were also conducted according to international guidelines in compliance with the ARRIVE guidelines, and approved by the Institutional Animal Care and Animal Experimentation Committee of the IGTP and the Government of Catalonia (DAAM5873 and DAAM6959). A total of 11 male rats, weighing 200–250 g, housed as stated above, were used in this part of the study. Transient stroke was performed under isoflurane anesthesia (4% for induction and 2% for maintenance) in a 30% O_2_/70% N_2_O mixture in Wistar rats (Charles River, Barcelona, Spain) by 90-min occlusion of the middle cerebral artery (MCAO) by intraluminal filament following the procedure described elsewhere [17]. After surgery, rats were placed on a heating pad at 37 °C, supervised until full recovery and then returned to their home cages. Two animals died before experiment termination, which was done by decapitation under isoflurane anesthesia. Brain samples were obtained 2 h after reperfusion and saved for WB, or for IP procedures followed by WB. For comparisons, the contralateral hemisphere of each animal served as control of the lesioned hemisphere. Experiments were carried out at lights on between 8 h and 20 h.

### 2.10. Statistical Analysis

Results are expressed and represented as the mean and SEM. Statistical analyses were performed using GraphPad Prism 8. Normality testing was carried out using the D’Agostino-Pearson omnibus K2 normality test (the Shapiro–Wilk or the Kolmogorov-Smirnov normality tests were used instead when required). Original data, or log-transformed data when necessary and useful to achieve homogeneity of variances, were analyzed using independent measures or paired Student’s *t*-test, or one-way ANOVA followed by the post-hoc Student-Newman-Keuls (SNK) test, as required, or Mann-Whitney U test when data were not gaussian or showed heteroscedasticity of variances. The effects were considered statistically significant at *p* < 0.05.

### 2.11. Antibodies

The following primary antibodies were used: goat anti-GAIP interacting protein at C terminus (GIPC) (1:250, sc-25658, RRID:AB_640993) and mouse anti-Ca^2+^/calmodulin-dependent protein kinase II (CAMKII) (1:200, sc-32288, RRID:AB_634551) were from Santa Cruz Biotechnology; mouse anti-phospho-DAPK1 at Ser308 (pDAPK1) (clone DKPS308, 1:50, D4941, RRID:AB_476906), rabbit anti-DAPK1 (1:400, D1319, RRID:AB_1078622), mouse anti-DAPK1 (clone DAPK-55, 1:400, D2178, RRID:AB_259206), rabbit anti-actin (1:500, A2066, RRID:AB_476693) and mouse anti-alpha-tubulin (1:5000, T6074, RRID:AB_477582) were from Sigma-Aldrich; anti-NMDAR 2B subunit (against the N-terminal extracellular region) (1:100, Millipore, MAB5782, RRID:AB_827428), rabbit anti-LRRFIP1 (1:400, Aviva Systems Biology, ARP38506_P050, RRID:AB_10570968); rabbit IgG isotype control (10500C, AB_2532981) and mouse IgG1 isotype control (MOPC-21) (MA1–10407, RRID:AB_2536775) were from Thermo Fisher Scientific; rabbit anti-glutathione peroxidase 4 (GPX4) (1:1000, ab125066, RRID:AB_10973901) was from Abcam.

The following secondary antibodies were used: for WB, IRDye-680 donkey anti-mouse (1:15,000, 926–32,222, RRID:AB_621844) and IRDye-800 donkey anti-rabbit (1:15,000, 926–32,213, RRID:AB_621848) were from LI-COR Biosciences; for immunocytochemistry, donkey anti-rabbit coupled to alexa fluor 488 (1:500, R37118, RRID:AB_2556546), donkey anti-rabbit coupled to alexa fluor 555 (1:500, A-31572, RRID:AB_162543), donkey anti-mouse coupled to alexa fluor 647 (1:500, A-31571, RRID:AB_162542) were from Thermo Fisher Scientific; donkey anti-mouse coupled to Cy3 (1:250, AP192C, RRID:AB_92642) was from Millipore.

## 3. Results

### 3.1. Effects of OGD on Neuronal Death, and on DAPK1 Expression, Dephosphorylation, and Cleavage

OGD-induced neuronal death was significant at 1 h, but not at 15 min, after reinstatement of normal oxygen and glucose (‘reperfusion’, Figure 1A). Based on this time-course, in order to detect changes in proteins preceding cell death, samples were obtained 30 min after OGD termination. Neurons exposed to OGD showed increased DAPK1 levels (Figure 1B), as assessed both by WB (considering immunoreactivity of all DAPK1 bands in Figure 1C) and ICC (Figure 1E), a 70% reduction of phosphorylated DAPK1 at Ser308 (pDAPK1) (Figure 1F), and a 22% reduction of the full length 160 kDa DAPK1 form that matched the increase in the truncated 100 kDa form (Figure 1D). We observed similar changes in the levels of total and truncated DAPK1 forms in brain tissue samples obtained 2 h after reperfusion from a rat transient MCAO stroke model (Figure 1B–D). Cleaved truncated DAPK1 forms have been previously reported after ischemia [11].

### 3.2. Main Effects of OGD on the Neuronal DAPK1 Interactome

Given the quantitatively different pattern of DAPK1 cleavage between control and OGD-exposed neurons, as explained above (Figure 1C,D), DAPK1 was first immunoprecipitated (IP) from a 1:1 mixture of control and OGD samples, using a specific rabbit anti-DAPK1 antibody against an N-term epitope that immunoprecipitates both the full length and the truncated DAPK1 forms; this antibody demonstrated specific when tested in DAPK1 knock-out samples [8]. No DAPK1 was immunoprecipitated when using a non-specific (NS) control IgG (Figure 2A). We did not observe significant differences in the amount of total DAPK1 pulled down in control or OGD samples (Figure 2B,C). WB showed that the bulk of the IP-DAPK1 in the OGD samples was in the truncated form (Figure 2B). To perform LC-MS protein identification of DAPK1 interactors, IP samples from control or OGD samples were eluted from the IP beads in a TCA solution, digested, and identified. A total of 596 proteins were identified in the DAPK1 interactome of control samples (Appendix A); in addition, a set of 116 proteins were only present and identified in the OGD samples (Appendix A). LC-MS differential proteomic screening provides a list of proteins that appear, disappear, or show a fold change > 1.4 or <0.6 when comparing the DAPK1 interactome of OGD neurons vs that of control ones. We found 190 proteins overrepresented (Appendix A) and 192 underrepresented (Appendix A) in the DAPK1 interactome after OGD. We used Ingenuity Pathway Analysis™ (IPA 2019, Qiagen, Germantown, MD, USA) to obtain insightful data analysis to identify proteins of our study that had not been previously reported as DAPK1 interactors (Figure 2D), and to test the effectiveness of our pulling down strategy for the study of the DAPK1/CTD-NR2B interactome. Peptide sequences specific for NR2B (Grin2b gene) or for the NR2B-interacting proteins GIPC and CaMKII were found increased among those identified by LC-MS in the DAPK1 interactome of OGD samples, as compared with control samples. Proteins of the DAPK1 interactome of control neurons, and those that appeared/increased their presence after OGD, were grouped under four criteria (molecular function, biological process, protein class, and cellular component) and classified within categories using Panther™ 15.0 (http://www.pantherdb.org) [18], and are shown as pie charts in Figure 3. In brief, the main represented classes (shown here with an added superscript number that matches that in Figure 3 for easy identification) within each category were: 1/molecular function: binding^1^, catalytic^2^, and structural^5^; 2/biological process: cellular process^5^, cellular component organization or biogenesis^4^, metabolism^11^, and biological regulation^2^; 3/protein class: cytoskeleton^5^, translation^14^, metabolite interconversion enzyme^8^, nucleic acid binding^9^, protein modifying enzyme^10^, and membrane traffic^7^; and 4/cellular component: cell^3^, cell part^2^, organelle^10^, organelle part^9^, and protein-containing complex^11^.

### 3.3. Several DAPK1 Interactors Are CTD-NR2B Binding Partners

The effectiveness of our IP-pulling down strategy in the study of the DAPK1 interactome, with especial relevance to CTD-NR2B, was further validated by WB. An increased presence of NR2B, GIPC and CaMKII were found in the DAPK1 interactome of neurons exposed to OGD (Figure 4A,B). The DAPK1 interactome contains the full length of 170 kDa NR2B subunit as the predominant form in control conditions, and mostly the truncated 115 kDa NR2B form in OGD conditions (Figure 4A). This is in accordance with the fact that NR2B is known to be cleaved during ischemia by calpain to render this 115 kDa form [19]. In vivo, protein extracts obtained from ipsilateral (ipsi, ischemic) brain hemisphere showed a 40% decrease of the full length of the 170 kDa NR2B band, together with a 50% increase of the 115 kDa NR2B band, as compared to the contralateral brain hemisphere (Figure 4C,D).

### 3.4. The New Protein Candidate LRRFIP1 Is Part of the DAPK1 Protein Complex during Ischemia In Vivo and Excitotoxicity In Vitro

The vast majority of proteins differentially present in the DAPK1 interactome of OGD-exposed neurons were identified for the first time as putative DAPK1 partners (based on Pubmed searches and in the fact that they are not listed in IPA hits as DAPK1 partners). Among them, we selected LRRFIP1 as a good candidate since it had been previously reported overexpressed in astrocytes exposed to ischemia [20]. OGD increased by 50% the ratio LRRFIP1/DAPK1 in the Co-IP-DAPK1 protein complex in cultured rat neurons (Figure 5A). We found that neurons express LRRFIP1 protein variants with the predicted MW (e.g., 83, 71 and around 50 kDa) [20] and that OGD increased by more than 50% the expression of both the 83 and the 50 kDa LRRFIP1 bands (Figure 5B,C).

In cultured rat neurons (Figure 6A,B), in either control, OGD or NMDA conditions, and ischemic rat brain homogenates (Figure 6C), we found that a DAPK1 specific antibody co-immunoprecipitates LRRFIP1. In turn, a LRRFIP1 specific antibody successfully co-immunoprecipitates DAPK1 in protein complexes, giving additional support to an interaction of the two molecules in a complex.

### 3.5. Colocalization of NR2B and DAPK1 with LRRFIP1

In control conditions, we observed by ICC that LRRFIP1 and NR2B are strongly associated at the neurites, and at the soma as well, but after treatment with NMDA, LRRFIP1 and NR2B concentrated mostly at the neuronal soma (Figure 7A). Regarding DAPK1, colocalization with LRRFIP1 was observed in the cytosol of neurons showing early traits of ischemic cell death (e.g., pyknotic nucleus), but not in neurons with normal morphology (e.g., large normal-looking nucleus) (Figure 7B).

### 3.6. The Ferroptosis Inducer Erastin Increases Neuronal LRRFIP1 Levels Preceding Loss of Neuronal Viability

OGD- and NMDA-induced cell death have been reported to have a ferroptotic component [21,22], and either cell death induced by OGD, by NMDA or by ferroptosis-pure-inducers such erastin have been associated to oxidative stress. We determined the effect of 10–20 µM erastin, an slow inducer of ferroptotic cell death, on LRRFIP1 in cortical neurons. Erastin, induced a concentration and time-dependent reduction of neuronal viability. A significant reduction in neuronal viability, as measured by a reduction of MTT, was observed at 24 h for the higher (20 µM) erastin concentration, whereas the lower concentration (10 µM) reduced cell viability at later time points (72 h) (Figure 8A). Twenty-four hours of treatment with erastin increased by 20% and 50% the 83 and 71 kDa LRRFIP1 bands, respectively (Figure 8B,C). Thus, WB results indicate that the lower erastin concentration (10 µM) increases LRRFIP1 levels preceding any significant effect on cell death. Moreover, ICC showed increased immunoreactive LRRFIP1 24 h after treatment with 20 µM erastin (Figure 8D).

## 4. Discussion

DAPK1 has been reported to be a main actor in the highly complex intracellular hubs driving neurodegeneration and/or survival organized around the intracellular NR2B-CTD signaling through CTD-CaMKII [23,24], CTD-PSD95-nNOS [7], and CTD-DAPK1 itself [8]. The body of knowledge of the last one is still scarce, probably because it was the last to be introduced, and further complicated by, at least, the following facts: 1/OGD increases DAPK1 levels preceding induction of neuronal death (Figure 1A–C), 2/OGD induces DAPK1 activation by dephosphorylation (Figure 1F) [8,14], and 3/proteolytic processing of DAPK1 by cathepsin around aa sequence 836–947 [11,25] generates 110 and 120 kDa bands, and a reduction of the full length DAPK1 form (Figure 1C,D). A previous study reported an OGD-induced increase of the full-length DAPK band in WB in vitro using a neuroblastoma cell line and longer exposure times to OGD [26]. These striking different conditions might account for the discrepancy. The cathepsin cleavage splits apart the DAPK1 kinase domain, located at the N-term and required to bind NR2B, from the canonical death domain located at the C-term.

To gain knowledge of new NR2B-CTD-related DAPK1 targets suitable to reduce excitotoxicity/ischemic-like damage, we used an affinity purification-mass spectrometry identification strategy. We used a specific and immunoprecipitation-qualified anti-DAPK1 antibody recognizing epitopes at the N-term, in the vicinity of the hotspot amino acid residues nearby the DAPK1 kinase domain [27]; as previously mentioned, this antibody demonstrated specific when tested in DAPK1 knock-out samples [8]. We found that our IP strategy resulted in full-length and truncated DAPK1 molecules. This was, in fact, observed by the OGD-induced increase in the relative levels of the DAPK1 100 kDa WB band, as compared with the full length DAPK1 molecule (160 kDa) pulled down in control samples (Figure 2B), as previously reported [11]. Truncated DAPK1 lacked a large fragment containing part of the cytoskeleton-binding region plus a phosphatase-binding site and the death domain.

We identified a total of 596 proteins interacting with the pulled down DAPK1 complex in control neurons (Appendix A; some of these proteins are depicted in Figure 2D). A previous report, using IP with anti-NR2B antibodies on previously fractioned synaptic and extrasynaptic pools and then analyzed using proteomics, obtained similar figures (700 proteins) [28]. LC-MS differential proteomics allowed us to screen and identify proteins in the DAPK1 interactome that consistently appeared, disappeared, or showed >1.4-fold change in OGD-exposed as compared to control neurons (190 were found overrepresented and 192 underrepresented; see Appendix A, respectively). Figure 3 shows a detailed Panther™ classification of all DAPK1 protein interactors in control neurons and the increased interactors in OGD-exposed neurons. We observed that several cytoskeleton proteins change their degree of interaction in the DAPK1 interactome after OGD. In general terms, some microtubule- and actin-related molecules, and also phosphatases, are underrepresented or lacking, whereas other cytoskeleton proteins such as tubulins, myosins, tropomyosins and molecules related to intermediate filaments tend to be increased. These changes are consistent with the fact that DAPK1 cleavage truncates the molecule into two halves of similar length, each half retaining half of the cytoskeleton-binding region. In this regard, the DAPK1 N-term half pulled down by IP lacks the phosphatase-binding site and the death domain. It is also noteworthy mentioning that, as far as we are aware, ours is the first study reporting a protein-protein interaction between DAPK1 and several histones, and that these physical interactions weaken after OGD. This is interesting since it has been described that histones may play a role in orchestrating the neuronal transcriptional response to experimental stroke by MCAO, OGD [29,30,31], and other challenges to cell survival such as excitotoxicity or ferroptosis [32,33,34]. Notably, we have also found that peroxiredoxin (PRDX) 1, PRDX2 and PRDX3, members of a family of antioxidant enzymes of crucial physiological importance, interact with DAPK1 and that this degree of association decreases after OGD.

As expected by the reported increased association of DAPK1 to NR2B during ischemia [8], our IP plus LC results show that OGD increased NR2B in the neuronal DAPK1 interactome (Figure 2D) and confirmed by IP plus WB (Figure 4A,B). Additionally, levels of other known CTD-NR2B-interacting molecules increased after OGD, including 1/PSD95, a.k.a. discs, large homolog 4 (DLG4) [4], 2/dedicator of cytokinesis protein 3 (DOCK3) [35], 3/protein kinase C [36], 4/CaMKII (confirmed by IP plus WB, Figure 4A,B) [23], and 5/NR2B-specific binding molecule GIPC (confirmed by IP plus WB, Figure 4A,B), this being a PDZ scaffolding protein reported to preferentially stabilize NR2B-rich NMDA receptors [37]. We observed these changes in the DAPK1 interactome despite 30% of NR2B molecules undergo ischemia-induced truncation to give rise to a 115 kDa WB band (Figure 4C,D), in agreement with previous reports [19,38,39]. In fact, most of the NR2B pulled down in the multiprotein complex in OGD-exposed samples is the truncated 115 kDa NR2B form, previously reported to lack a portion of the C-term intracellular tail due to calpain proteolytic activity boosted by ischemia or glutamate [39,40,41].

The role of DAPK1 in ischemic excitotoxic damage has been so far associated to postsynaptic mechanisms of neurodegeneration. Nonetheless, very recent data suggest that DAPK1 might serve different roles at different locations of the neuron, since the DAPK1 interactor caytaxin, which we have also identified in the proteome of our DAPK1 Co-IP strategy (Appendix A), has been reported to inhibit the catalytic activity of DAPK1 at presynaptic sites and to exert neuroprotective effects in stroke [42].

As one of the aims of the study was to find novel DAPK1 protein interactors differentially increased in the neuronal DAPK1 interactome after OGD, we focused on leucine-rich repeat of flightless I-interacting protein 1 (LRRFIP1), a.k.a. GC-binding factor 2 (GCF2), for further study due to several reasons. First, other proteins, previously identified as binding partners of LRRFIP1 in other reports, e.g., the molecules LRRFIP2 and flightless I homologue (FLII) [43,44], are also enriched in the OGD-DAPK1 interactome in our study. Second, and most important, LRRFIP1 has been previously observed upregulated in ischemia models, either in vivo or in cultured astrocytes [20,45]. In the rat, five transcripts have been identified so far that show upregulation in ischemia models [20,45]. These transcripts give rise to protein isoforms with predicted molecular masses of 83, 71, 48.9, 46.1 and 44.9, the last three isoforms displayed as a unique WB band of around 50 kDa. Furthermore, it has been reported that a prevalent polymorphism in the promoter of the glutamate transporter EAAT2 gene creates a new consensus binding site for LRRFIP1, which impairs glutamate uptake in astrocytes and increases the frequency of early neurological worsening in stroke [45]. Our findings in the DAPK1 interactome were further confirmed by the LRRFIP1/DAPK1 Co-IP ratio, which showed an increase of more than 50% (Figure 5A). Moreover, WB showed that OGD increased the expression of the major 83 and 50 kDa LRRFIP1 bands in neurons (Figure 5B,C). As far as we are aware, this is the first study reporting a direct LRRFIP1 interaction with DAPK1. In fact, several cross-IP assays (IP with antibodies against DAPK1 or LRRFIP1) revealed an association and coexistence of the 2 proteins after ischemia in adult rat brain or after OGD or NMDA treatment in primary culture neurons (Figure 6). Our results further suggest that LRRFIP1 might be playing an important role in ischemic brain damage and in OGD-treated neurons exposed to high oxidative stress conditions. In this regard, the potent natural antioxidant/inhibitor of oxidative stress resveratrol has been reported to increase LRRFIP1 levels in human peripheral blood mononuclear cells [46,47].

Beyond the increase in the expression of LRRFIP during OGD, inspection by confocal microscopy revealed that LRRFIP1 and NR2B colocalize at neurites, and also cell bodies, in control neurons, whereas both molecules are mostly evident at the soma after a brief NMDAR overactivation period (30 min) (Figure 7A) preceding excitotoxicity, a similar relocation effect that has also been observed for NR2B in previous reports [39,48,49]. Associated with this relocation of LRRFIP1, we also observed colocalization of LRRFIP1 with DAPK1 in the cell body of neurons undergoing nuclear pyknosis or fragmentation, but not in neurons shortly exposed to NMDA and showing normal morphology or in control neurons (Figure 7B). Our results might indicate a rearrangement within the neuron of LRRFIP1 molecules associated to NR2B or DAPK1 preceding excitotoxic neuronal death.

To further extend these results in order to know whether LRRFIP1 might be involved in other ROS-promoting neuronal slower death-inducing paradigms such as ferroptosis, we tested the ferroptosis inducer erastin in neuronal primary cultures. We observed that erastin (at concentrations that increase membrane lipid peroxides, not shown) increased levels of LRRFIP1 83 and 71 kDa bands, as measured by WB (Figure 8B,C), preceding neuronal death (Figure 8A); ICC showed increased total immunoreactive LRRFIP1 at 24 h at a higher erastin concentration (Figure 8D). Erastin acts mainly by inhibiting the cystine/glutamate antiporter system Xc- (composed of SLC7A11 and SLC3A2) [50,51], leading to cysteine starvation, glutathione depletion and deleterious excesses of cellular ROS. We observed that a 24-h exposure to the high 20 µM erastin concentration was enough to induce a significant increase of ferroptotic neuronal death, whereas longer exposure (72 h) was required to produce significant cell death at a lower erastin concentration (10 µM) (Figure 8A). We observed this increase in LRRFIP1 in mature neuron cultures at 24 h after 10–20 µM erastin with no change in levels of GPX4. A recent study reports erastin-induced reduction of cell viability associated to reduced levels of GPX4 [52], but the authors use very different conditions: immature 5 DIV neurons, exposed to higher erastin concentration (50 µM) and for 48 h. As impaired Xc- activity results in depletion of intracellular glutathione, an essential cofactor of GPX4, erastin would be expected to impair the enzymatic activity of GPX4, with a concomitant increase in ROS accumulation, even with no changes in protein GPX4 levels, as we have found (Figure 8B,C).

To date, the biological activity and physiological role of truncated forms of DAPK1 and NR2B, and of LRRFIP1 transcripts are not fully understood. LRRFIP1 has been reported in human and mouse with distinct roles in transcriptional repression, modulation of the innate immune response, regulation of cell division with an impact in cancer cells or wound healing, and as a modifier of platelet function and thrombosis [53]. Additionally, some previous results observed increased expression in brain tissue or astrocytes exposed to ischemia [20]. Our results suggest that the increase in LRRFIP1 expression precedes neuronal death as a common characteristic of ROS-generating challenges, involving either fast (ischemia/ischemia-like and excitotoxicity) or slow (ferroptosis) mechanisms of neuronal death.

## 5. Conclusions

We have identified proteins up and down-regulated in the neuronal DAPK1-interactome by OGD, some of them related to NMDAR and validated by WB. We have identified for the first time LRRFIP1 as a DAPK1 partner, which we found up-regulated by OGD in the neuronal DAPK1 interactome, this being in line with the previously reported involvement of LRFFIP1 in ischemia. LRRFIP1 levels increased in neurons exposed to pro-oxidant conditions such as purely ferroptotic-induced cell death by exposure to erastin or to OGD-induced cell death, in which an excitotoxic and a ferroptotic-induced cell death component can converge. Therefore, the present study identifies LRRFIP1 as a novel partner of DAPK1, suitable as a target for neuroprotection following ischemic challenges.

## Figures and Tables

**Figure 1 antioxidants-09-01202-f001:**
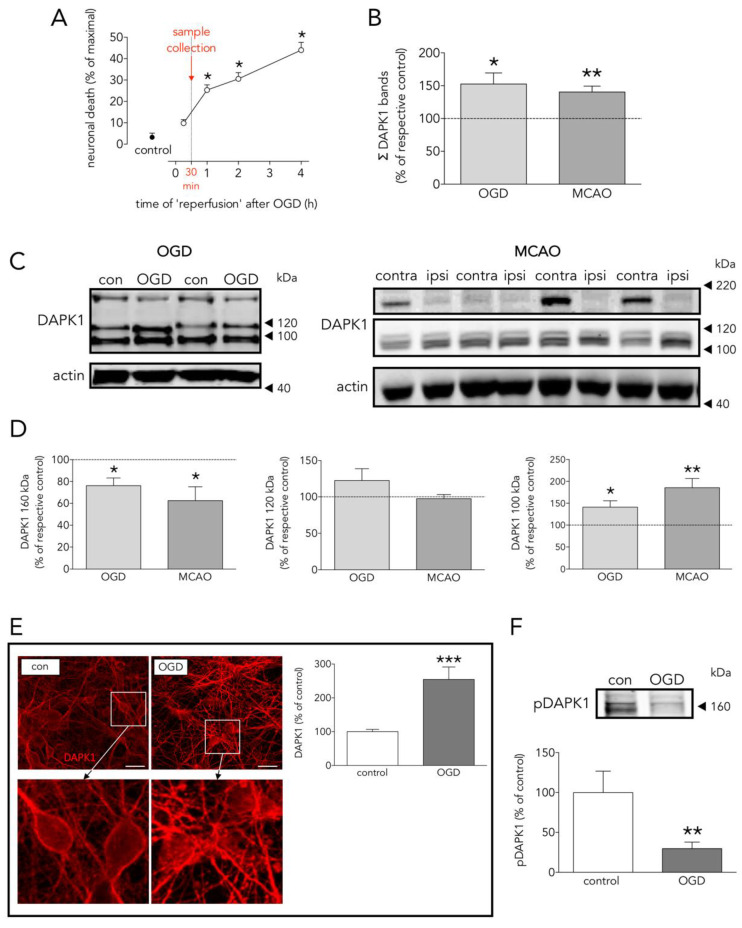
Effects of OGD on neuronal death, and DAPK1 expression, cleavage, and dephosphorylation. (**A**) Effect of time of ‘reperfusion’ after OGD on neuronal death; as depicted, experimental samples were taken at 30 min for further processing (one-way ANOVA plus SNK, *n* = at least 5 independent primary neuron culture preparations, with at least 3 technical replicates each). (**B**) Effect of OGD and MCAO on total DAPK1 levels (OGD: *t* test, *n* = 3 independent primary neuron culture preparations, with 3 technical replicates each; MCAO: paired *t* test, *n* = 9 rats, comparing ipsilateral (ipsi, ischemic) brain hemisphere with the contralateral (contra, control) one). (**C**) WB showing the effect of OGD or MCAO on DAPK1 bands. (**D**) Quantification of the effect of OGD or MCAO on DAPK1 cleavage and levels of the resulting bands (OGD: *t* test, *n* = 3 independent primary neuron culture preparations, with 3 technical replicates each; MCAO: paired *t* test, *n* = 9 rats, comparing ipsilateral (ipsi, ischemic) brain hemisphere with the contralateral (contra, control) one). (**E**) Effect of OGD on DAPK1 levels as measured by immunocytochemistry; insets in the upper pictures are shown magnified in the bottom pictures (*t* test, *n* = 3 independent primary neuron culture preparations, with 5 technical replicates each). (**F**) WB and quantification of the effect of OGD on pDAPK1 levels (*t* test, *n* = 3 independent primary neuron culture preparations, with 3 technical replicates each). Results are shown as the mean and SEM. * *p* < 0.05, ** *p* < 0.01, *** *p* < 0.005 vs. respective control in all graphs. Scale bar: 30 µm. Note that in (**B**), total DAPK1 levels represent protein expression.

**Figure 2 antioxidants-09-01202-f002:**
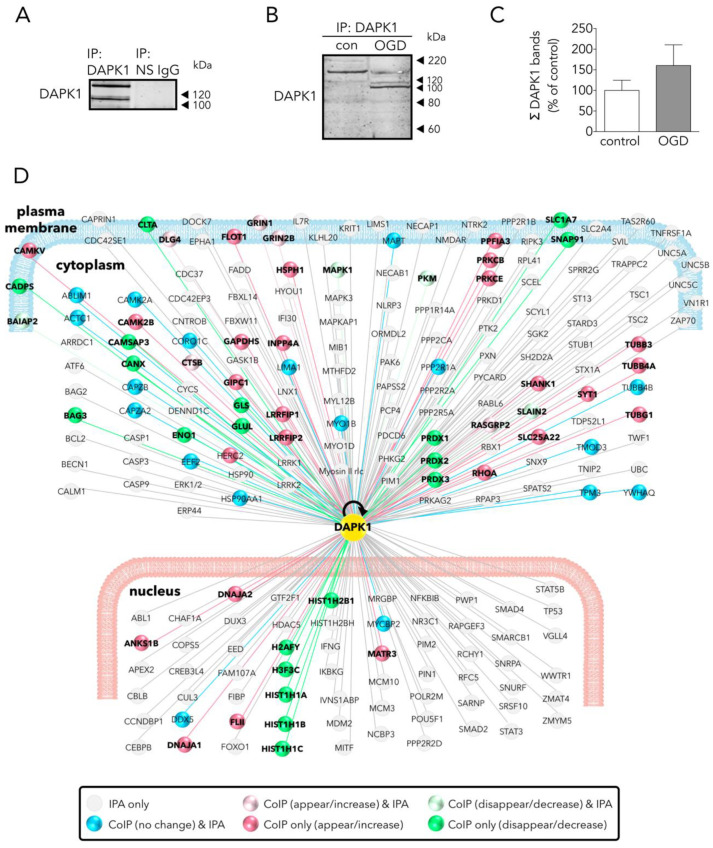
Main effects of OGD on the neuronal DAPK1 interactome. (**A**) WB shows that DAPK1 immunoprecipitates using a specific anti-DAPK1 antibody (IP: DAPK1, left lane), but it does not when using a non-specific antibody (IP: NS IgG, right lane). (**B**) WB showing the effect of OGD on the DAPK1 bands obtained by immunoprecipitation. (**C**) Graph showing the amount of DAPK1 obtained after immunoprecipitation of control and OGD samples (*t* test, *n* = 3 independent primary neuron culture preparations, with 3 technical replicates each). Note that, in this graph, differently than in WB in Figure 1B, DAPK1 data represent protein co-immunoprecipitated from neurons. (**D**) Drawing depicting some of the DAPK1 interactors found in the present study and those present in the IPA database; color codes of proteins are self-explained in the legend within the drawing. Results are shown as the mean and SEM.

**Figure 3 antioxidants-09-01202-f003:**
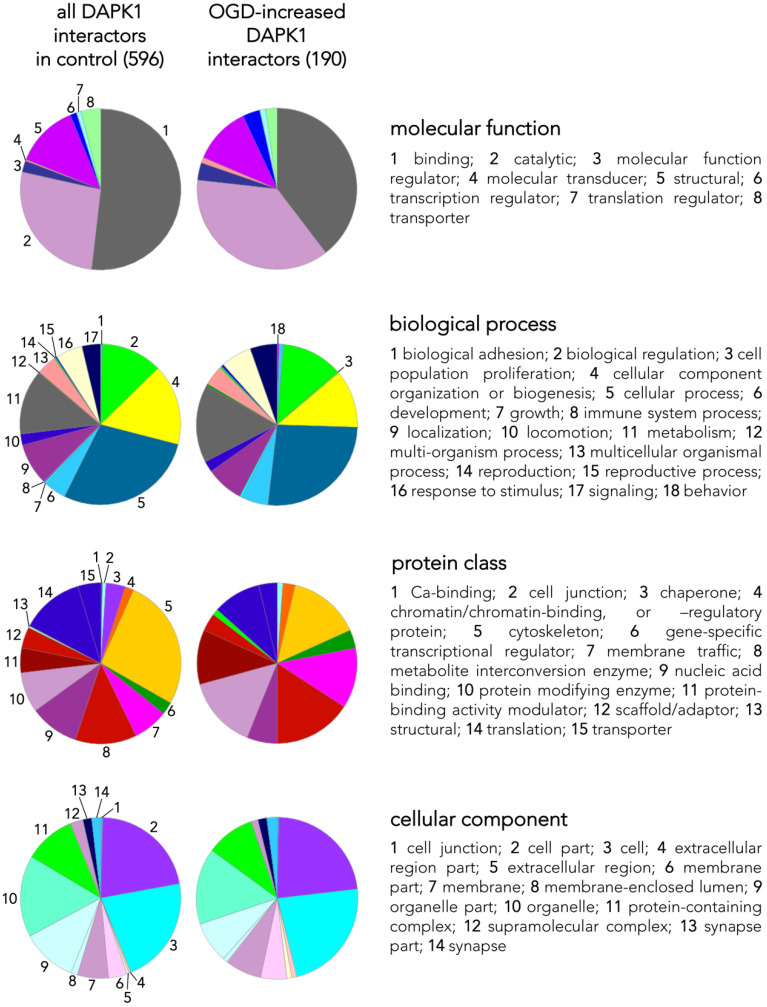
Classes and categorization of proteins in the DAPK1 interactome. Panther™ v15.0 (pantherdb.org) was used to classify proteins of the DAPK1 interactome in control neurons and those interactors found increased in OGD-exposed neurons (they are shown in a pie chart on the left). Four main classes are defined, each encompassing a number of different categories encoded by colors and explained on the right side of each class.

**Figure 4 antioxidants-09-01202-f004:**
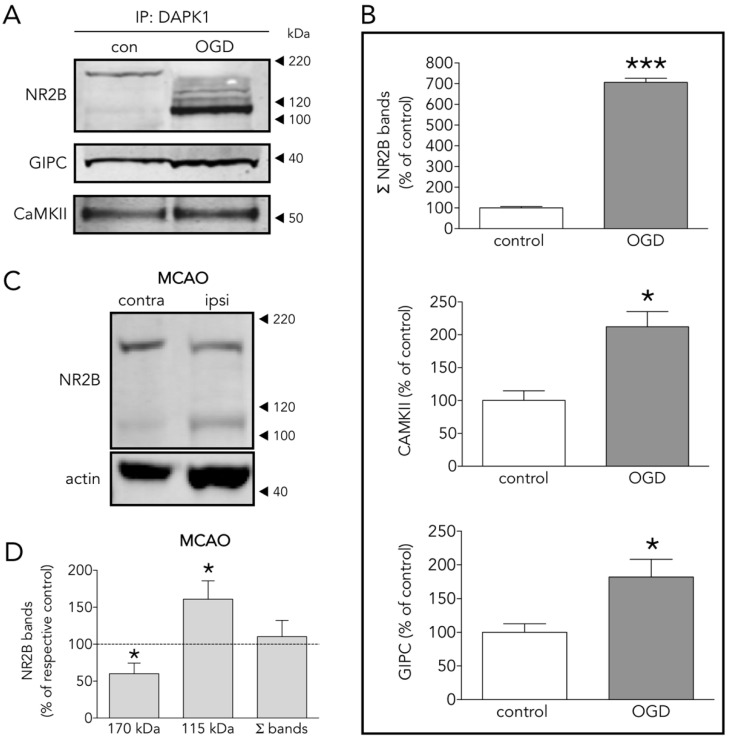
Several DAPK1 interactors are CTD-NR2B binding partners. (**A**) WB and (**B**) quantification of the effect of OGD on the co-immunoprecipitation of DAPK1 with the well-known DAPK1 interactors NR2B, CAMKII and GIPC (*t* test, *n* = 4–5 co-immunoprecipitations from independent primary neuron culture preparations). (**C**) WB showing the effect of MCAO on the NR2B band levels. (**D**) Effect of MCAO on 170 kDa, 115 kDa and total NR2B levels (paired *t* test, *n* = 9 rats, comparing ipsilateral (ipsi, ischemic) brain hemisphere with the contralateral (contra, control) one). The results are shown as the mean and SEM. * *p* < 0.05, *** *p* < 0.005 vs. respective control in all graphs.

**Figure 5 antioxidants-09-01202-f005:**
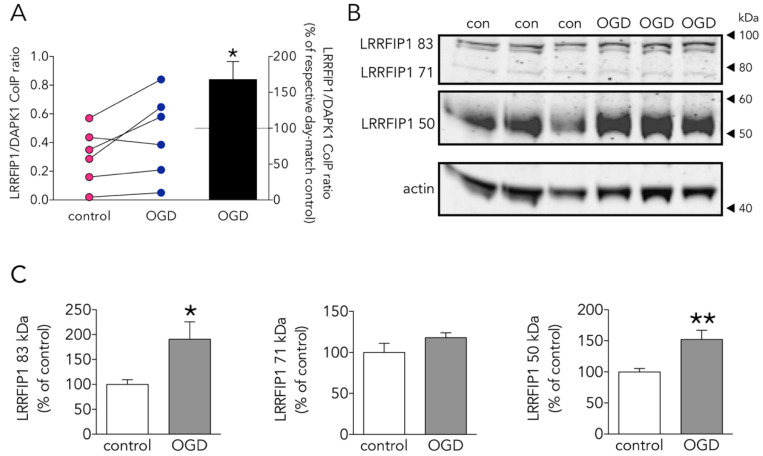
Effect of OGD on the new DAPK1-interacting candidate protein LRRFIP1. (**A**) Effect of OGD on the LRRFIP1/DAPK1 co-immunoprecipitation ratio (*t* test, *n* = 6 co-immunoprecipitations from independent primary neuron culture preparations). (**B**) WB showing the effect of OGD on the expression of different LRRFIP1 transcripts, and (**C**) graphs showing the quantification of this effect (83 and 50 kDa graphs: *t* test, and 71 kDa graph: Mann-Whitney U test, *n* = 3 independent primary neuron culture preparations, with 3 technical replicates each). Results in A (left panel) are shown as individual co-immunoprecipitation values, and in A (right panel) and C as the mean and SEM. * *p* < 0.05, ** *p* < 0.01 vs. respective control in all graphs.

**Figure 6 antioxidants-09-01202-f006:**
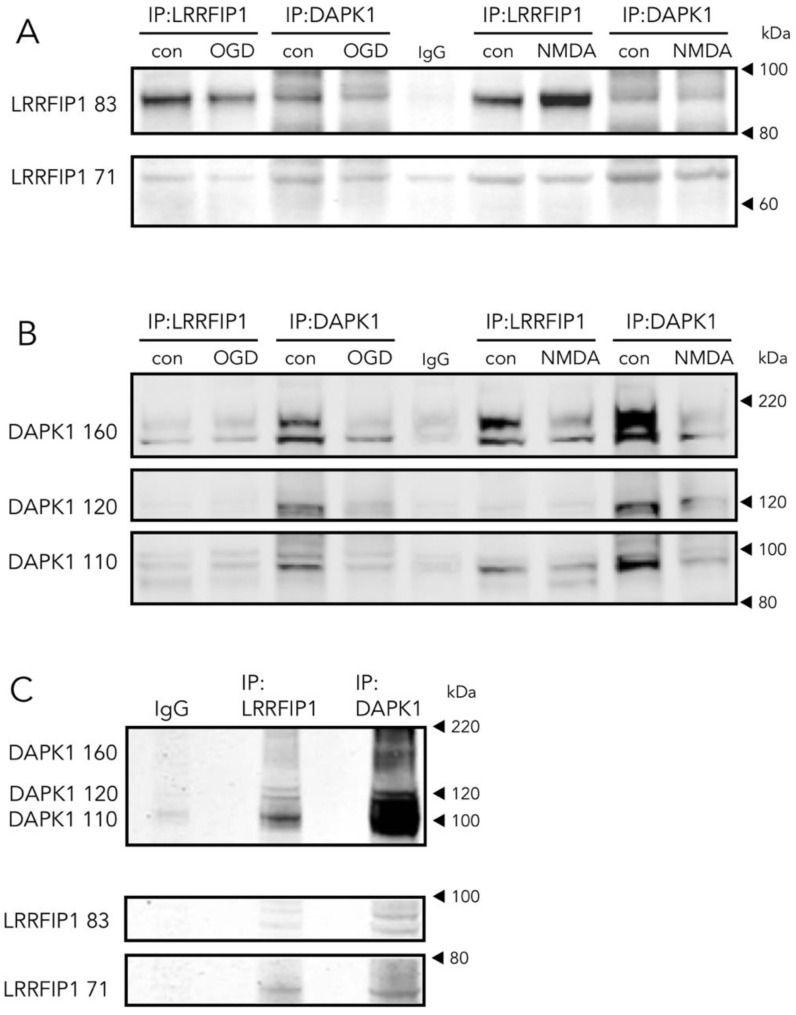
Direct and reverse co-immunoprecipitation of LRRFIP1 during ischemia in vivo and excitotoxicity in vitro**.** (**A**,**B**) WB showing the effect of OGD or NMDA on direct and reverse co-immunoprecipitation of LRRFIP1 and DAPK1 from rat cultured neurons. (**C**) WB showing direct and reverse co-immunoprecipitation of DAPK1 and LRRFIP1 from ischemic rat brain homogenates.

**Figure 7 antioxidants-09-01202-f007:**
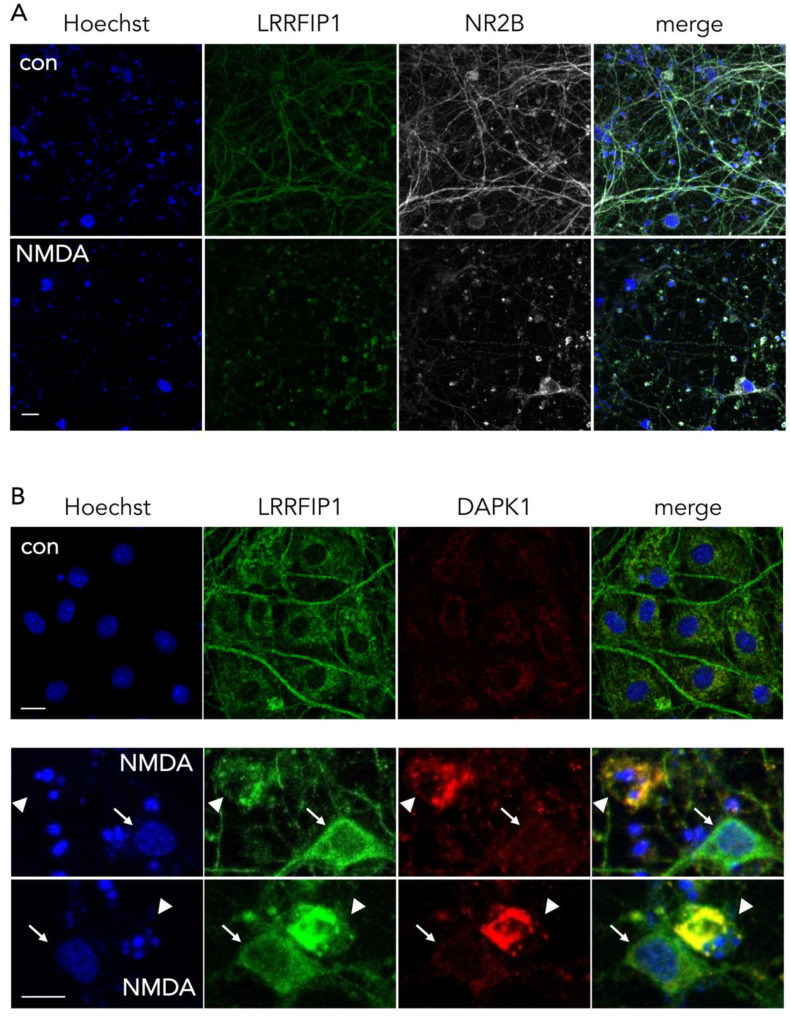
Representative ICC images showing the colocalization pattern of NR2B and DAPK1 with LRRFIP1. (**A**) Images showing the main cellular location of LRRFIP1 and NR2B in control and NMDA-treated neurons in vitro; scale bar: 30 µm. (**B**) Images showing the colocalization pattern of LRRFIP1 and DAPK1 in control neurons (upper panel, con), and in neurons with normal (arrows) or pyknotic (arrowheads) nucleus shortly after NMDA treatment (lower panels, NMDA); scale bars: 10 µm.

**Figure 8 antioxidants-09-01202-f008:**
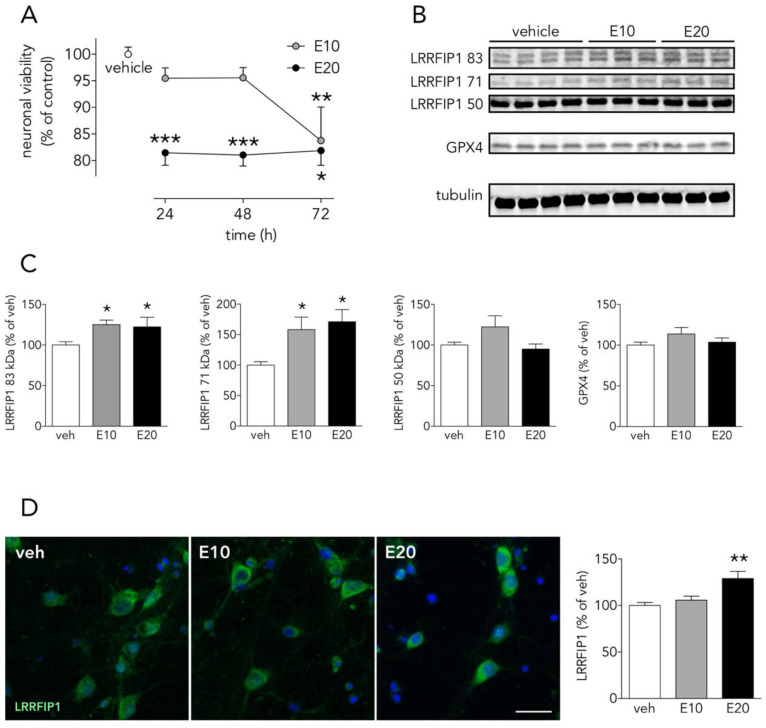
The ferroptosis inducer erastin increases neuronal LRRFIP1 levels. (**A**) Time-course effect of erastin on neuronal viability in vitro (one-way ANOVAs plus SNK, *n* = 3 independent primary neuron culture preparations, with 3 technical replicates each). (**B**) WB showing the effect of erastin on LRRFIP1 and GXP4 levels at 24 h, and (**C**) graphs depicting the quantification of this effect (one-way ANOVA plus SNK, *n* = 3 independent primary neuron culture preparations, with 3 technical replicates each). (**D**) Representative ICC images showing LRRFIP1 expression in neurons exposed to vehicle (veh) or erastin for 24 h, and a graph showing the quantification of this effect. Ten µM erastin: filled grey circles/bars or E10; 20 µM erastin: filled black circles/bars or E20. Results are shown as the mean and SEM. * *p* < 0.05, ** *p* < 0.01, *** *p* < 0.001 vs. vehicle in all graphs. Scale bar: 20 µm.

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
