# Peer review of "Comparative Proteomics Unveils LRRFIP1 as a New Player in the DAPK1 Interactome of Neurons Exposed to Oxygen and Glucose Deprivation"

_antioxidants, 2020, doi:10.3390/antiox9121202_

Round 1
Reviewer 1 Report
This study uses a discovery platform to identify dynamic protein networks associated with death-associated protein kinase 1 (DAPK1) in rat cortical neurons treated with an oxygen/glucose deprivation (OGD) to model redox perturbations. Data suggests identification of a novel interaction with LRRFIP1 which was validated by co-IP and further examined during OGD and ferroptosis. DAPK1 protein networks could inform on a multitude on redox and excitotoxic responses in nervous tissues as well as other physiologic responses since DAPK1 is highly expressed in other tissues. However, there were major technical issues based on authentication of antibodies used for detection and immunoprecipitation of DAPK1 which need clarification or further experimentation.
Major concerns
- I can’t identify any literature describing full-length and truncated DAPK1 and none is cited. Therefore, it’s difficult to convince me that the three immunoreactive bands in Fig 1C represent different sizes of DAPK1 as opposed to cross-reactivity with an unknown peptide. Were genetic (e.g., siRNA knockdown, overexpression, DAPK1 knockout samples) experiments performed to confirm DAPK1 specificity? Also, I cannot figure out how data in Fig 1B were quantified since independent DAPK1-reactive bands were not upregulated at 150% compared to controls individually (Fig 1D) yet the collective data shows this. Is reporting total DAPK1 even relevant if there are cleavage-specific changes?
- Similar concern regarding antibody specificity for Fig 2A. PMID20141836 is cited to demonstrate specificity since they probe brains from DAPK1 wt, hets, and KO mice. However, they only showed a single immunoreactive band ~160 kDa, suggestive of full-length DAPK1. This could suggest species-specific differences (mice v rat), age-dependent DAPK1 cleavage, and/or antibody detection discrepancies. This again highlights the importance of using the aforementioned DAPK1 genetic controls. Without these controls demonstrating specificity, it is very difficult to glean anything from the proteomic data.
- Why do anti-DAPK1 antibodies used in Fig 1 & 2 give different banding patterns in control samples even though there is detection of supposed “truncated” DAPK1 bands?
- Unclear in methods, results, and data if IPs were performed with technical and/or biological replicates. Would have expected to see some language about peptide presence across multiple samples.
- Same issue of antibody specificity for IP-IB experiments in Fig 6B/C – starting to see even more DAPK1-immunoreactive bands even though these are immunopurified samples.
Minor concerns
- Were there any differences in regard to sex?
Reviewer 2 Report
In the present manuscript entitled: “Comparative proteomics unveils LRRFIP1 as a new 2 player in the DAPK1 interactome of neurons exposed to oxygen and glucose deprivation”, the authors, using two ischemia-reperfusion models (oxygen and glucose deprivation (OGD) and transient occlusion of the middle cerebral artery(MCAO) in rats) and combining IP with affinity purification-mass spectrometry, try to identify new proteins of the DAPk1 interactome, which can serve as intervention targets of excitotoxicity/ischemic-like damage. The study describes the LRRF1P1 as a new mediator of the NR2B-DAPK1 complex in ischemia, however, the results fail to provide conclusive evidence.
Main concerns:
- According to the graph in Fig. 5A, OGD stimulates LRRFIP1/DAPk1 co-IP ratio but the WBs in Fig. 6 don’t show that increase.
- In Fig. 7B colocalization between LRRFIP1 and DAPk1 fluorescence is only shown in cells exposed to NMDA, while control cells are missing. Intriguingly, and although a statistical evaluation of the confocal microscopy images is needed, at first sight, colocalization is observed exclusively in cells with high MAPk1 staining. Colocalization between LRRFIP1 and DAPk1 should be analyzed in control cells, and confirm that DAPk1 red fluorescence does not overlap the green channel.
- Ferroptosis has been associated with OGD and NMDA-induced cell death. Using erastin as an inducer, the authors try to link ferroptosis with changes in LRRFIP1 levels, however, and unlike what is stated in the text, neither the WB nor the ICC images in Fig 8 show a clear increase in LRRFIP1, even at low doses of erastin.
- Transient occlusion of the middle cerebral artery (MCAO) in rats is used as an in vivo stroke model. Graphs in Fig 1D display an enhanced cleavage of DAPk1 in MCAO samples, in line with results from neurons exposed to OGD, however, the WB in Fig. 1C does not reflect these changes compared to control samples. Also, the 50% increase of the 115 kDa NR2B band after MCAO described by the authors is hard to observe in the WB displayed in Fig 4C. To avoid these discrepancies, WBs in both figures should be replaced by more representative ones.
- In Fig. 1F, the graph should be accompanied by a representative WB.
- In line 449, the following sentence: "the three death cell forms..." is confusing, what types of cell death are the authors referring to?
Reviewer 3 Report
This is an exciting and impressive study in which the authors claim to have identified LRRFIP1 as a novel differential DAPK1 partner. Moreover, LRRFIP1 levels were increased by pro-oxidant conditions such as ODG or the ferroptosis inducer erastin. Using high-resolution LC-MS/MS, they also identified the interactome of DAPK1, which comprises more than 600 proteins involving binding, catalytic, and structural molecular functions. In general, the experiments are well-conducted and the methodological approach is appropriate. I think this is an exciting and novel contribution that will be of broad interest to the readers of the journal. I do not have any major concerns about the work, but the manuscript could be improved by addressing the following issues:
- In Figures 1E and 7A, the quality of representative images is relatively low. A lower magnification can improve the quality of images significantly.
- In Figure 8D, it is better to have a quantification for the images to show the difference between each group.
Reviewer 4 Report
In this study, DeGregorio-Rocasolano and colleagues investigated DAPK1 interactome using neuronal cells exposed to OGD. DAPK1, which is known to be associated with NMDAR, seems to undergo proteolytic cleavage in response to OGD, leading to the reduction in the levels of full-length DAPK1, as well as phosphorylated DAPK1. Using control and OGD-treated cells immunoprecipitated with DAPK1, the author performed LC-MS/MS analysis to identify DAPK1-interating proteins. Then, the authors focused on LRRFIP1 proteins as an important interacting protein, and further demonstrated that interaction and colocalization between DAPK1 and LRRFIP1 increased upon NMDA treatment. Overall, understanding the molecular mechanism of DAPK1 in neuronal cell death is an important issue and this study provide a valid information on various DAPK1-interacting proteins. However, there are several techinical issues that should be addressed as follow.
- OGD seems to decrease the expression of full-DAPK1 and pDAPK1, possibly suggesting that OGD inhibit the kinase activity of DAPK1. Could you provide any evidence on the activity of DAPK1 by detecting the phosphorylation levels of DAPK1 target genes or by kinase assay?
- The representive western data for pDAPK1 in Fig. 1F should be provided.
- In Fig. 5B, there are large flucutations in actin levels between two control lanes, making it difficult to determine the exact expression levels of actin and LRRIP1 isoforms. Therefore, this experiment should be repeated.
- In Fig. 6, I have some concerens about the non-specificity of immunoprecipitation because there are similar bands between IgG and IP lanes.
- In Fig. 7B, do LRRFIP1 and DAPK1 also colocalize in normal condition?
- In Fig. 8, western blot data did not support meaningful alteration in protein levels of LRRFIP1s, and immunofluoresence experiments were also also quantitative. Also, even with slight changes in protein levels, current data did not provide any potential relevance of LRRFIP1 in ferroptosis. Therefore, this reviewer recommends removing the data related to ferroptosis.
Round 2
Reviewer 2 Report
The authors have adequately answered all of my concerns. The article has considerably improved. I recommend it for publication.
Reviewer 4 Report
First of all, I did not know that this manuscript had already been evaulated by three reviewer.
Although my concerns have not been fully resolved, the authors have provided reasonable explanations, which I also agree with them.
Therefore, I support the publicantion of the current manuscript.